# Theranostic Applications of 2D Graphene-Based Materials for Solid Tumors Treatment

**DOI:** 10.3390/nano13162380

**Published:** 2023-08-20

**Authors:** Daniela Iannazzo, Consuelo Celesti, Salvatore V. Giofrè, Roberta Ettari, Alessandra Bitto

**Affiliations:** 1Department of Engineering, University of Messina, 98166 Messina, Italy; ccelesti@unime.it; 2Department of Chemical, Biological, Pharmaceutical and Environmental Chemistry, University of Messina, 98165 Messina, Italy; sgiofre@unime.it (S.V.G.); rettari@unime.it (R.E.); 3Department of Clinical and Experimental Medicine, University of Messina, 98125 Messina, Italy; alessandra.bitto@unime.it

**Keywords:** graphene-based materials, solid tumors, drug delivery, theranostic tools

## Abstract

Solid tumors are a leading cause of cancer-related deaths globally, being characterized by rapid tumor growth and local and distant metastases. The failures encountered in cancer treatment are mainly related to the complicated biology of the tumor microenvironment. Nanoparticles-based (NPs) approaches have shown the potential to overcome the limitations caused by the pathophysiological features of solid cancers, enabling the development of multifunctional systems for cancer diagnosis and therapy and allowing effective inhibition of tumor growth. Among the different classes of NPs, 2D graphene-based nanomaterials (GBNs), due to their outstanding chemical and physical properties, easy surface multi-functionalization, near-infrared (NIR) light absorption and tunable biocompatibility, represent ideal nanoplatforms for the development of theranostic tools for the treatment of solid tumors. Here, we reviewed the most recent advances related to the synthesis of nano-systems based on graphene, graphene oxide (GO), reduced graphene oxide (rGO), and graphene quantum dots (GQDs), for the development of theranostic NPs to be used for photoacoustic imaging-guided photothermal–chemotherapy, photothermal (PTT) and photodynamic therapy (PDT), applied to solid tumors destruction. The advantages in using these nano-systems are here discussed for each class of GBNs, taking into consideration the different chemical properties and possibility of multi-functionalization, as well as biodistribution and toxicity aspects that represent a key challenge for their translation into clinical use.

## 1. Introduction

Despite the outstanding advances in cancer management, leading to improved survival degree, the death rate of patients affected with solid tumors, often caused by metastasis, has not changed significantly during the last decades [1]. The main reason for the encountered failure is due to obstacles in pharmacokinetics, including whole-body, intra-tumoral and cellular pharmacokinetics [2], involving the administration of optimized drug delivery pathways [3] or the use of anticancer drugs assisted by cell permeation enhancers [4]. Intratumor therapy has additional advantages in the treatment of solid tumors, allowing the administration of high doses of anticancer agents within the tumor, avoiding drug resistance phenomena and systemic toxicity [5]. Therefore, extensive efforts have been focused on advanced drug delivery systems, designed to be directly injected into tumors under direct visualization or imaging guidance [5]. The challenge in these therapeutic approaches is related to the complicated biology of the tumor microenvironment. The presence of a perforated endothelium which leads to enhanced permeation and retention (EPR) effect, intra-tumoral poor vascularity, partially or completely blocked lymphatic drainage, intra-tumoral hypoxia and acidosis are the major reasons for solid tumors’ treatment failure [6]. 

Solid tumors are constituted by two distinct and interdependent cellular entities: (1) the epithelial parenchyma formed by the malignant cells and (2) the stroma, which represents the supporting connective tissue where the abnormal cells are dispersed [7]. The stroma plays an important role in regulating the development and homeostasis of normal tissues but also creates niches that promote metastatic cancer cell seeding [8]. In addition, the cancer stromal cells contribute to the development of drug resistance phenomena by creating a physical barrier that prevents intra-tumoral drug penetration and secreting cytokines and chemokines that confer resistance to chemotherapeutic treatments [9]. The extracellular matrix (ECM) of tumor cells is biochemically distinct in its composition from the normal ECM; the higher stiffness increases the interstitial fluid pressure which in turn, interferes with the effective diffusion of anticancer agents within the solid tumor. Blood vessels formed during angiogenesis, the early step in tumorigenesis, are highly irregular, presenting tortuous and leaky vessels with abnormal wall structure and heterogeneous blood flow [10]. This heterogeneity leads to interstitial hypertension and impaired blood supply, thereby impeding the efficient delivery of therapeutic agents to the tumors. Different techniques in combination with conventional chemotherapeutics have so far been proposed for solid cancer ablation in various body organs, such as microwave (MW) dynamic or thermal therapy, and ultrasound treatments [11,12]. MW assisted methods have been explored for tumor ablation, due to the deep penetration of MW in tissues, high heating efficiency, and negligible side effects. However, the used MW energy (10^−3^ eV) is deemed insufficient to induce the generation of free radicals [11]. Ultrasound treatment can also offer unique opportunities to optimize therapy but also requires combined use with chemotherapeutic agents [12]. The nanoparticles (NPs) based approach has shown the potential to overcome these limitations by enabling transport through gaps between endothelial cells in the tumor blood vessels, formed during angiogenesis [13]. The pathophysiological features of solid cancers, characterized by leaky vasculature and compromised lymphatic drainage, facilitate the passive or ligand-mediated active tumor targeting of these NPs, thus allowing the delivery of therapeutic agents inside the tumors (Figure 1) [14]. Moreover, nanotherapeutics have shown the ability to ameliorate tumor hypoxia, a characteristic feature of most solid tumors related to angiogenesis, and to efficiently inhibit tumor growth [15]. Once the NPs accumulate from the blood to the tumor site, they can pass cell membranes and enter cells by endocytosis. Nano-scaled carriers, endowed with great surface-to-volume ratio, can allow controlled and triggered drug release [16], improve the water dispersibility of poorly soluble chemotherapeutics and prolong their circulation in the physiological milieu [17].

Among the different classes of nanomaterials designed for nanomedicine, 2D graphene-based nanomaterials (GBNs) such as functionalized graphene, graphene oxide (GO), reduced GO (rGO) and graphene quantum dots (GQDs), the last generation of 2D graphene fragments, have attracted great interest in biomedical research, including for drug delivery, biosensing, bioimaging, and photothermal therapy (PTT) [18,19,20]. The fascinating connection between these nanomaterials and medicine originates mainly from their small sizes (typically in the range of 1 nm–1 µm) which allow improved transport through inter-endothelial junctions, and also from their similar dimensions to biomolecules whose functions are fundamental for life, such as proteins and DNA [21,22]. However, their interest goes beyond their size and is related to their unique chemical, physical and optical properties, near-infrared (NIR) light absorption, tunable biocompatibility and versatile surface functionalization, which allow their contemporary conjugation with different biopolymers, drugs, cancer targeting ligands and bioimaging agents [23,24,25,26]. Their high photothermal conversion efficiency under NIR light irradiation, due to the presence of the graphene structure, has been exploited for the development of theranostic nanoparticles for photoacoustic imaging-guided photothermal–chemotherapy and for PTT and photodynamic therapy (PDT) applied to tumor destruction [27]. The GBN family includes also GBN-based nanocomposites containing inorganic and/or organic components [18]. Their functionalization by covalent or non-covalent approaches can improve their biocompatibility, achieve efficient drug loading, and ability to cross cells and tissue barriers [19,25]. These nanomaterials, in fact, can be contemporary modified with different biomolecules such as polysaccharides, proteins, and nucleic acids, thus allowing targeted cancer therapy [24,28]. In addition to drug delivery applications, GBNs have demonstrated great potential in cancer detection, thus paving the way for new integrated approaches that combine diagnosis and therapy [28,29]. These theranostic applications can be achieved using photosensitizers (PSs), so allowing PDT, or using heating agents under NIR light for PTT [30,31,32]. PTT, in combination also with PDT, represents an emerging method for solid tumors’ treatment, allowing their ablation using NIR light in the range of 700–1000 nm [30]. These strategies are possible due to the unique features of graphene structure whose ideal band gap (0.3–0.7) allows efficient NIR light emission and heat conversion [31]. Studies in vitro and in vivo have reported promising results for GBNs functionalized with different biomolecules, drugs and targeting ligands, demonstrating the synergistic effect of PDT/ PTT treatments that use light to kill cancer cells through the generation of reactive oxygen species (ROS), and/or heat [33] (Figure 2).

In this review, we report the recent advances related to the development of the main classes of 2D GBNs, namely graphene, GO, rGO and GQDs, as innovative nanoplatforms for the development of nano-formulations to be used for the diagnosis and treatment of solid tumors. The advantages of using these nanomaterials and the different applications such as drug and gene delivery, bioimaging, PTT and PDT for solid tumor treatment will be discussed for each class of GBNs, taking into consideration their chemical functionalization, biodistribution, and toxicity aspects, which represent a fundamental challenge for their translation in clinics.

## 2. Theranostic Applications of 2D GBNs for Solid Tumors’ Treatment

The theranostic applications of GBNs for treating solid cancers are strongly dependent on the shape, size, hydrophilic/lipophilic behavior, kind and degree of functionalization of this family’s members, which include mainly graphene, GO, rGO, and GQDs. These factors can greatly influence their application in the biomedical field. In principle, materials to be used for cancer therapy should be stable at physiological conditions, biocompatible, endowed with high affinity with biological substrates, able to cross cell membranes and easily cleared by physiological systems [34]. Due to its unique and fascinating chemical and physical properties, high surface area and versatile functionalization, graphene is one of the most promising materials for biosensing, tissue engineering and electronic applications, while the more oxidated derivatives, such as GO, rGO and GQDs, have proven to be superior for nanomedicine applications, due to their excellent hydrophilicity and stability in the physiological environment [35]. GO, consists of single-layer sheets of sp^2^ hybridized carbons and, because of the presence of oxygen-containing functionalities, is highly hydrophilic and dispersible in water-based solvents. The presence of these functionalities and the large surface area allows interaction with various small molecules, macromolecules and polymers via π-π stacking, covalent bonding, hydrophobic interactions, electrostatic forces, and hydrogen bonding, highlighting its potential in nanomedicine [31]. The chemical reduction of GO removes most of the oxygen functional groups and partially restores the sp^2^ carbon bonds of graphene layers, giving rGO. This nanomaterial is more conductive and hydrophobic than GO, enabling interaction with hydrophobic anticancer drugs and small molecules via π-π stacking; moreover, due to its high light-absorbing capacity, rGO is a promising material for theranostic applications [24,27]. GQDs, the next generation of GBNs, are fragments, limited in size, of 2D graphene sheets with a diameter less than 20 nm in lateral dimension and single or few layers of thickness. Owing to their very small size, GQDs exhibit the quantum size effect and are endowed with strong size-dependent photoluminescence properties. Analogously to the other GBNs, GQDs have NIR light absorption properties and can be conjugated with biomolecules, polymers, inorganic NPs and labelling agents for the simultaneous treatment and diagnosis of cancer [20]. The surface modification of GBNs by covalent and non-covalent functionalization strategies have been shown to greatly improve drug delivery performance [36]. The insertion of biocompatible polymers can increase the stability of GBN-based nanocarriers while the use of recognition molecules can improve their targeting ability; moreover, their combination with other NPs, can produce a synergic theranostic effect [37]. GBN-based nanocomposites have demonstrated the potential to overcome problems related to conventional delivery systems, further addressing their application in cancer therapy [38]. The most important members of the GBN family will be discussed, reporting the more recent advances in their applications as a theranostic tools for the treatment of solid cancers.

### 2.1. Theranostic Tools Based on Graphene

The high surface area of graphene sheets allows its biofunctionalization with different organic and inorganic substrates by covalent and non-covalent strategies [39]. Metal and metal oxide (MOx) NPs can be loaded onto the graphene surface to give stable graphene/MOx nanostructures with improved biological performance [40]. Metal–organic compounds have attracted great attention, due to their potential in theranostics [41,42]. Hatamie and co-workers functionalized graphene sheets with cobalt-citric acid to develop graphene/magnetic metal composites, as magnetic fluid hyperthermia (MFH) and contrast agents in magnetic resonance imaging (MRI) [43]. The synthesized nanomaterials were found to be biocompatible, as evaluated in vitro against L929 mouse fibroblasts cells, and showed a superior conversion of electromagnetic energy into heat at 350 kHz frequency. MRI results also showed that these graphene/cobalt nanocomposites are promising MRI contrast agents. Boron dipyrromethene (BODIPY) is a fluorescent dye widely investigated in the biomedical field for cell imaging, and cancer photo-theranostics, due to its high light chemical stability [44,45]. Meng and co-workers exploited BODIPY as photosensitizer for the development of a graphene-based phototheranostic agent using mitomycin C as anticancer drug [46]. Biological tests, performed in vitro in HeLa cells, demonstrated good ROS production ability, high photothermal conversion efficiency (48%), and excellent anticancer activity. Moreover, superior fluorescence and photothermal imaging ability in the agar model was also reported, thus demonstrating the great potential of this nanomaterial for fluorescence/photothermal dual-model imaging. Several studies demonstrated the ability of graphene-based nanocomposites to be used as effective theranostic agents for treating solid cancers [47,48,49] (Table 1).

A theranostic nano-system based on graphene for the treatment of unresected and chemo-resistant ovarian cancer cells was developed by Tarantula and co-workers using an intraoperative multimodal phototherapy [47]. The authors modified graphene nanosheets (GN) containing low amounts of oxygenated functionalities, with poly-propylenimine (PPI) dendrimers carrying the photosensitizer phthalocyanine (Pc). The system was also functionalized with polyethylene glycol (PEG) to enhance the biocompatibility of the system (GN-PEG) and to link the peptide LH-RH, which is involved in the synthesis and release of luteinizing hormone. In this study, low-power NIR irradiation of single wavelength was applied for heat generation and ROS production. The so combined PDT/PTT therapy produced the destruction of LHRH receptor-positive A2780/AD multidrug resistant human ovarian carcinoma cells, with a killing efficacy of 90–95%. When tested in an animal model of human ovarian carcinoma xenografts from nude mice, this nano-system confirmed its potential as an effective NIR theranostic probe for imaging and combinatorial phototherapy. Graphene nanoflakes (GNFs), constituted by graphene sheets of about 30 nm in diameter and edge-terminated with carboxylic acid functionalities, have been also investigated as substrates for the development of theranostic agents. Holland and co-workers functionalized GNFs with a peptide able to bind prostate-specific membrane antigen (PSMA), as targeting ligand, the anti-mitotic drug (R)-ispinesib (R-Isp), the chelating agent desferrioxamine B (DesB), and with an albumin binding tag to extend the in vivo half-life [48]. In vitro biological tests performed on human prostate adenocarcinoma cells (LNCaP) demonstrated the ability of the system to induce G2/M phase cell cycle arrest, with high specificity toward the PSMA expressing cells. The authors also labelled the system with the radioactive diagnostic agent gallium-68 (^68^Ga) and evaluate its suitability in theranostics by performing experiments in vivo in athymic nude mice bearing subcutaneous LNCaP tumors. Time–activity profiles obtained from dynamic positron-emission tomography (PET) demonstrated low accumulation and retention in background tissue and rapid renal clearance. Graphene nanoribbons (GNRs), strips of graphene with dimension in width lower than 100 nm, also demonstrated significant photothermal conversion efficiencies [49]. He and co-workers, investigated these nanomaterials as PPT agents for the triple negative breast cancer (TNBC), using as targeting ligand mannose or galactose [50]. The system was supra-molecularly ensembled with a pyrene-tagged peptide ligand (PRGD), able to bind the α_ν_β_3_ integrin receptors, to afford a dual-receptor targeting function. PTT tests performed in vitro against the MDA-MB-231 cell line and in vivo, by tail vein injection in tumor bearing live mice, demonstrated almost complete tumor ablation and no tumor growth after 15 days, thus clearly demonstrating the potential of this dual-receptor targeting for effective cancer therapy.

### 2.2. Theranostic Tools Based on Graphene Oxide

Compared to graphene, GO possesses a more hydrophilic nature, due to the presence of many oxygen-containing functional groups on the basal plane and on the edges. The presence of these functionalities allows the formation of hydrogen bonds in water, thus greatly improving its suitability for applications in nanomedicine [51]. Analogously to graphene, this nanomaterial can convert the NIR light energy into thermal energy and, when employed for cancer therapy, the produced hyperthermia, results ultimately in the thermal elimination of cancer cells [52,53,54]. Its outstanding loading ability, due to the large surface area, makes GO a multifunctional nanoplatform for PTT/PDT combination therapy. Some studies report the development of GO-based nanocomposites, containing Gadolinium ion (Gd^3+^), a contrast agent for magnetic resonance imaging (MRI) [55]. A bimodal theranostic nano-delivery system, based on GO and containing anticancer agents and gadolinium and gold nanoparticles for MRI, have been synthesized, demonstrating in vitro the suitability of the system as chemotherapeutic and diagnostic agent [56,57]. Theranostic systems based on GO and containing molecules with cancer targeting abilities, such as folic acid (FA) or hyaluronic acid (HA), have been also investigated for chemo–PTT and PDT applied to solid tumors [58,59,60,61,62]. Hybrid composites based on GO and iron oxide NPs have been successfully employed for magnetic hyperthermia therapy [63,64]. These systems, loaded with anticancer agents, demonstrated low cytotoxicity in vitro, and enhanced anticancer activity when compared to the free drugs. Zarrabi and co-workers combined the imaging abilities of magnetic nanoparticles with chitosan-grafted GO [65]. This pH-sensitive nano-carrier, loaded with doxorubicin (DOX), demonstrated high biocompatibility when tested in a healthy L929 cell line (derived from mouse connective tissue) and also increased T2 contrast efficacy after grafting high molecular weight chitosan, due to the better surface coverage. Many recent papers have reported the development of GO-based nanocomposites for the treatment of deep-seated or large solid tumors [66,67,68,69,70,71,72,73,74,75]. Some recent and relevant studies on the use of GO-based theranostic tools for solid cancers treatment are reported in Table 2 and will be discussed here. Zhou and co-workers synthesized a composite based on GO and chitosan (GO-CS) for the treatment of melanoma, the most malignant body surface tumor, using a cellular penetrating peptide (MPG) as effective transmembrane drug carrier and microRNA (miRNA), as regulator of melanoma cell glycolysis [67]. In this study, GO obtained from acidic oxidation of graphite was covalently conjugated with CS and MPG and then loaded with miRNA. The in vitro experiments, performed on melanoma A375 cells, demonstrated the ability of the system to inhibit the cell growth by a synergistic action of MPG and miRNA. The results of in vivo studies, performed on nude mice, by subcutaneous tumor implantation experiments have shown a strong reduction of tumor volumes after 35 days, thus demonstrating the applicability of the system for melanoma therapy. The nano-system was demonstrated also to be a safe and efficient gene nanocarrier. A theranostic tool for cancer imaging, PTT/PDT, and NIR photothermal therapy was synthesized by Li and co-workers using a GO-based nanocomposite conjugated with the PS phthalocyanine, through a silicon moiety (SiPc) [68]. The nano-system showed intrinsic fluorescence, synchronous PTT/PDT effect, and higher NIR absorbance when compared to GO. Biological tests, performed in vitro in human breast cancer cell (MCF-7) and human cervical cancer cell (HeLa), demonstrated the intracellular fluorescence of the system, the ROS generation and the effective photoablation of cancer cells triggered by NIR laser at 671 and 808 nm. The in vivo systemic administration in MCF-7 xenograft mice demonstrated that SiPc@GO can be accumulated in the tumor, inhibiting its growth after laser irradiation, along with satisfactory biocompatibility.

A magnetic GO-based nanocomposite for synergistic chemo/PTT effect was synthesized by Chen and co-workers by directly growing γ-Fe_2_O_3_ NPs on the surface of pegylated GO [69]. The system was loaded with the anticancer drug DOX and biologically evaluated through in vitro and in vivo experiments. The results of in vitro studies performed in Hela cells demonstrated the low cytotoxicity and cancer cell inhibition by the synergistic chemo and PTT effect, as well as increased drug release upon 808 nm NIR light irradiation. The tail vein injection of the system in hepatoma-22 (H22) tumor-bearing nude mice demonstrated its ability to be detected by T2-weighted magnetic resonance. In a similar study, a multifunctional theranostic system, based on GO and MnWO_4_, grown in situ on the GO-PEG surface, was loaded with DOX and investigated as PTT and photoacoustic imaging (PAI) agent [70]. The nanocomposite proved to be a good drug carrier when evaluated in vitro in human umbilical vein endothelial cells (HUVECs), also demonstrating the ability to release the drug at low pH and through NIR light irradiation. These results were also confirmed by in vivo experiments in 4T1-tumor female athymic nude mice, indicating the good synergistic anticancer effect of the combined chemo–PTT. Kim and co-workers reported the synthesis of palladium NPs decorated GO for PDT and PTT of prostate solid tumors [71]. In vitro studies performed against prostate cancer cells (PC3) showed a concentration-dependent toxicity following NIR irradiation. In vivo distribution studies carried out for 48 h in a PC3 xenograft mouse model, following intra-tumoral injection, demonstrated the high accumulation of the nano-system inside the tumor and minimal distribution in heart, lung, kidney, liver, and spleen. Moreover, upon NIR laser irradiation, an effective tumor volume reduction with minimal toxicity to other organs was reported, thus suggesting the suitability of this system for prostate cancers. A stimulus–responsive, biocompatible and biodegradable system with theranostic ability was synthesized by Selvaraj and co-workers using, as nanoplatform, a graphene oxide flake decorated liposomal (GOF–Lipo) nanohybrid [72]. In this study, red fluorescent GOFs were functionalized with folic acid (FA) as targeting ligand through the linker cysteamine, loaded with DOX, and added to phospholipid thin films. The subsequent lipid film hydration and extrusion processes allowed the obtaining of GOF–Lipo spheres with a size distribution of about 200 nm (Figure 3a). The nano hybrid demonstrated in vitro, against red blood cells, a good cell uptake, due to the binding of FA with the folate receptors expressed on the surface of cancer cells (Figure 3b) and a significant biocompatibility. The multi-stimuli (NIR light and pH) triggered response demonstrated, both in vitro and in vivo, the good theranostic performance of the system (Figure 3c,d). Experiments performed in vivo in 4T1 tumor bearing Balb/c mice demonstrated the specific biodistribution of the nanohybrid at the tumor site, and a major tumor regression (8.3%) when a single dose of the system was injected intravenously. A GO-based theranostic system containing polydopamine (PDA), bovine serum albumin (BSA), the contrast agent DTPA-Mn(II), FA as targeting agent, and 5-fluorouracil (5Fu) as anticancer drug, was developed by Shervedani and co-workers as drug delivery system for colon cancer and for MRI applications [73].

The in vitro experiments performed in CT-26 colon cells and histopathological tests demonstrated the anticancer ability of the system and its biocompatibility. MRI studies in vivo, using male Wistar rats, also allowed verification of the suitability of the system as diagnostic agent. A GO-based nano-theranostic platform using superparamagnetic iron oxide nanoparticles (SPIONs) grown in situ on GO nanosheets via a hydrothermal process, was synthesized by Chen and co-workers, for drug delivery and MRI applications [74]. The nano-system, covalently linked with DOX, showed in 4T1 cancer cells controllable pH-responsive anticancer activity. Moreover, when injected in mice bearing 4T1 tumor, high-resolution T1-weighted MRI performance and good anticancer activity was reported. A nano-system based on GO-PEG, modified with the epidermal growth factor receptor (EGFR) as targeting ligand [76], and loaded with 5-Fu, was developed by Hu and co-workers for the treatment of colorectal cancer (CRC) [75]. The nanocomposite showed good anticancer efficacy, both in vitro and in vivo. When evaluated in vitro against EGFR-over-expressing HCT-116 cells, effective drug delivery was observed; a good biocompatibility showing very low hemolysis rate (<10%) was also reported. The in vivo evaluation of the nano-system theranostic performance was performed in subcutaneous CRC bearing mouse model, by injecting HCT-116 cells into nude Balb/c mice. These studies demonstrated the ability of the system to accumulate at tumor site and a tumor growth inhibition rate of 90% after NIR treatment among all experimental groups, thus holding great promise for the treatment of CRC.

### 2.3. Theranostic Tools Based on Reduced Graphene Oxide

The reduced form of GO, rGO, proved to be an excellent photothermal agent, allowing for efficient in vivo tumor ablation, and also integrating imaging and therapy for cancer treatment [77]. The substantial difference, in terms of physicochemical properties, between these two GBNs lies in their synthesis, deriving rGO from the treatment of GO with reducing agents [78]. GO have a higher content of oxygen-containing moieties and, as a consequence, it is more wettable and soluble in water than rGO. Moreover, the presence of oxygenated moieties allows its conjugation with biomolecules and polymers, thus improving its use in drug delivery [79]. On the other hand, rGO is endowed with significantly higher electrical conductivity and photothermal absorption, due to the maintenance of the sp^2^ -hybridized carbon atoms structure, which is of particular interest for solid cancers’ treatment [80,81]. Multifunctional nano-systems based on rGO have shown great potential for the temperature/pH-dependent drug/gene delivery and for multimodal cancer therapy [82]. NIR light-responsive nanocomposites based on rGO have been developed by Jaiswal and co-workers for enhanced in vitro chemo–PTT [83]. In this study, the authors simultaneously reduced and functionalized GO, obtained from graphite powder, using poly(allylamine hydrochloride) and loaded the system with DOX. High biocompatibility, good drug-loading capacity and enhanced NIR absorbance were reported for this nano-system. In addition, in vitro studies performed in MCF-7 breast cancer cells demonstrated efficient drug delivery and a synergistic chemo–photothermal activity. A superparamagnetic nanohybrid based on Fe_3_O_4_ and rGO, exploited for the chemo-thermal therapy of human cervical cancer effect using DOX as model drug, demonstrated excellent cancer inhibition during magnetic field assisted hyperthermia treatment [84]. Nanocarriers based on rGO have been developed for the treatment of gliomas, the most common primary malignant tumors of the central nervous system [85]. Grodzik and co-workers demonstrated the ability of rGO to decrease the mitochondrial membrane potential and to increase the expression of the ki-67 gene in the glioblastoma cells of the multiform U87 cancer cell line [86]. In a subsequent study, the same research team reported that different types of GBNs may activate distinct cell pathways and that the cytotoxic activity of rGO in U87 glioma cells was related to the presence and types of oxygen-containing functional groups on the graphene surface [87]. Several rGO-based nanomaterials with NIR light absorption and heat transfer properties have been developed for the treatment of solid cancers [88,89,90,91,92,93,94] (Table 3). In order to amplify the effect of PA imaging and PTT, Cai and co-workers developed a photo-theranostic system based on indocyanine green-loaded polydopamine-rGO (ICG-PDA-rGO) by spontaneously self-polymerization on dopamine on rGO and chemical adsorption of the fluorescent dye [88]. The synthesized nano-system showed enhanced optical absorption in NIR wavelength and low toxicity. A significant improvement in the PA and PTT effects was reported both in vitro in 4T1 breast cancer cells and human BEAS-2B normal bronchial epithelial cells, and in vivo in 4T1 tumor-bearing mice, compared to those exhibited by the nano-systems lacking the dye. Gold nanorods (Au NRs) are known as effective photothermal contrast agents, both in vitro and in vivo [95]. Szunerits and co-workers synthesized PEG functionalized rGO (rGO-PEG), decorated with Au NRs (Au NRs@rGO-PEG), for the photothermal treatment of human glioblastoma astrocytoma (U87MG) cells in mice [89]. The authors covalently linked to the PEG chain the NIR dye cyanine-7 (Cy7) as fluorescent marker, and functionalized the nano-system with the Tat peptide, a targeting ligand for U87MG cells. When injected in U87MG tumor-bearing mice, high accumulation in the tumor of this multifunctional theranostic nanohybrid was observed. Due to the specific interaction of Tat peptide with cancer cells and after a low dose of NIR light excitation, a significant reduction in tumor size was observed after 5 days (irradiation at 0.7 W cm^−2^ for 10 min).

Wu and co-workers synthesized a nanoplatform for the synergistic targeted chemo-PTT using PDA functionalized rGO coated with mesoporous silica (MS), modified with HA as targeting ligand (PrGO@MS-HA), and then loaded with DOX [90]. The authors reported a pH-dependent and NIR laser irradiation triggered DOX release with enhanced chemo–PTT effect both in vitro in HeLa cells, exposing CD-44 receptors, and in vivo in BALB/c mice implanted subcutaneously with HeLa cells. To investigate the in vivo distribution, the authors loaded the nano-system with the fluorescent dye Cyanine5 (Cy5), and intravenously injected the system and the dye alone in different groups of mice (Figure 4). A strong fluorescence signal from the liver was recorded in all the treated groups which was attributed to the hepatic macrophage uptake. After 24 h, a strong fluorescence signal at the tumor site was recorded, thus highlighting the good targeting and retention ability of the nano-system (Figure 4a). The ex vivo fluorescence images of the tissues obtained at 24 h after injection showed a greater fluorescence for those treated with the nano-system with respect to those treated with Cy5 group, confirming the ability of the system to accumulate at the tumor site (Figure 4b). A weak fluorescence signal in both groups was reported in liver and kidney, probably due to renal clearance, while no fluorescence signals were recorded in heart, spleen, and lung. Pancreatic cancer is a malignant tumor with high mortality and gradually increasing incidence rate [96]. Li and co-workers investigated the anticancer activity of rGO, at different concentrations, combined with NIR laser (980 nm), in animal pancreatic cancer [91]. Experiments in vivo performed in female C57BL/6 mice implanted with Panc02-H7 pancreatic cancer cells showed higher treatment temperature and slower tumor growth using the higher laser dose (0.75 W/cm^2^) and the higher concentration of rGO (2 mg/kg), thus further highlighting the potential of this nanomaterial for pancreatic cancer treatment. Cancer immunotherapy has recently demonstrated to be a valuable alternative therapeutic strategy to fight cancer. However, the traditional immunotherapies are expensive and often lead to toxic side effects [97]. To overcome these issues, Chen and co-workers developed a hybrid nanocomposite for photothermal–immunotherapy, based on Fe_3_O_4_ NPs and rGO, which was pegylated (Fe_3_O_4_/rGO-PEG) to activate macrophages by triggering cytokine release [92]. The authors demonstrated in vitro in 4T1 cells, and in vivo in tumor-bearing BALB/c mice, immunogenic cancer cell death after laser irradiation (805 nm), reporting also the activation of dendritic cells in tumor draining lymph nodes.

When investigated in vivo, the nano-system demonstrated a photothermal effect leading to the destruction of primary tumors with a significant increase in the survival time of tumor-bearing animals. In addition, the potential of the system for MRI-guided photothermal immunotherapy for metastatic cancers was also highlighted. A synergistic anticancer immunity approach was also proposed by Ma and co-workers by developing a multifunctional system for PTT, inhibition of indoleamine-2,3-dioxygenase (IDO) and a block of programmed cell death-ligand 1 (PD-L1) [93]. In this study, rGO nanosheets were functionalized with PEG and folic acid FA as targeting ligand, and loaded with the IDO inhibitor epacadostat, via π−π stacking interaction. The rGO–PEG–FA system was also labelled with the NIR dye Cy7, to evaluate the cell uptake in CT26 cells. Experiments in vivo, performed in BALB/c mouse inoculated with CT26 cells, under NIR irradiation (808 nm, 1 W/cm^2^) demonstrated that the combined PTT, IDO inhibition, and PD-L1 block, were effective for inhibiting tumor growth of both irradiated cancers and those at distant sites, without PTT treatment. In a recent study, El-Zahed and co-workers reported a study on the in vivo toxicity and anticancer activity of a nanocomposite based on rGO and silver NPs (rGO/Ag), bio-synthesized using the crude metabolite of *Escherichia coli* [94]. The authors injected the nanocomposite in mice with Ehrlich ascites carcinoma (EAC) for 7 days at a dose of 10 mg/kg, observing a moderate toxicity and histopathological effects in kidney and liver. Antiproliferative effect on EAC cells, reduced ascites volume, and maintained mice survival was also reported.

### 2.4. Theranostic Tools Based on Graphene Quantum Dots

The next generation of GBNs, GQDs, are fragments, limited in size, of 2D graphene sheets. Due to their very small size, these nanomaterials embrace the favourable features of GBNs, namely chemical and physical stability, easy surface multi-functionalization and NIR light absorption, with the extraordinary properties derived from the quantum size confinement effect [98,99]. Due to the unique size-dependent effect, these nanomaterials exhibit stable strong size dependent photoluminescence, attracting great attention in optoelectronics and the biomedical field [100,101]. GQDs demonstrated a great potential in cancer research due to their inherent high and stable fluorescence, resistance to photobleaching, high colloidal stability in water and lower cytotoxicity when compared to the other members of the GBN family [102]. In addition, GQDs alone have proved themselves to be anticancer agents due to their ability to enter the cell nucleus and bind to DNA fragments [103,104]. Analogously to the other GBNs, the presence of more active groups on their surface allows their conjugation, with targeting ligands, drugs, biopolymers, inorganic NPs and labelling agents, making them ideal nanoplatforms for the simultaneous treatment and tracking of cancer cells [105,106,107,108,109]. Several studies in vitro report the use of GQDs as effective imaging and PTT/PDT agents [110,111,112,113]. Badrigilan and co-workers synthesized bismuth coated GQDs as theranostic nanoprobe for CT imaging and cancer PTT, reporting a photothermal efficiency of 30.0% in HeLa cells [110]. The same research group developed superparamagnetic iron oxide (SPIO) and bismuth oxide coated GQDs for in vitro computed tomography and magnetic resonance (CT/MR) dual-modal imaging and cancer-specific PTT, demonstrating the ability of the system for imaging-guided tumor therapy [111]. Naumov and co-workers reported a study on GQDs obtained from rGO or HA, as PTT and bioimaging agents [112]. When tested in vitro in HeLa cells, using 808 nm laser irradiation, a substantial photothermal heating together with a decrease in cell viability was recorded. Zheng and co-workers reported an interesting study related to the ability of GQDs, with different lateral sizes, to generate singlet oxygen ^1^O_2_ under light irradiation and their ability to enhance the photoactivity of PS such as methylene blue and methylene violet [113]. From this study, it emerged that GQDs with lateral sizes in the range of 5–20 nm were not able to generate ^1^O_2_ under 660 nm laser or halogen light, inhibiting also the photoactivity of the investigated photosensitizers. These results underline the central role played by the methods used for the synthesis of GQDs and by the kind of functionalization on the photoactivity of this class of nanomaterials. Different papers have investigated nano-systems based on GQDs for the treatment of solid cancers [103,114,115,116,117,118,119,120,121,122,123]. Some recent advances related to GQD theranostic tools for solid tumors are reported in Table 4 and will be discussed here.

Hu and co-workers developed a red blood cell (RBC) membrane enveloped nano-sponge as stealth agent and photolytic carrier able to transport docetaxel-loaded (DTX) GQDs inside cancer cells [115]. In this study, GQDs prepared from artificial graphite were loaded with the hydrophobic drug (DTX) via π−π interactions and encapsulated in silica spheres obtained from tetra-ethyl-orthosilicate (TEOS) hydrolysis, via a template-based mechanism. Then, the system was capped with RBC and with Cetuximab (Ct), a monoclonal antibody with targeting properties towards EGFR-overexpressing cells. In vitro biological studies demonstrated good uptake in EGFR-expressing cells (A549). The targeted delivery into tumor sites was investigated in vivo by injecting the nano-system into nude mice bearing A549 tumor cells, through the tail vein. A greater accumulation at the tumor site was reported for the system containing Ct with respect to the same system lacking the targeting antibody. The applied NIR irradiation was demonstrated to increase penetration and drug delivery into the tumor tissue causing damage and growth inhibition in 21 days, after a single irradiation. Moreover, after 56 days, no tumor recurrence was observed, and no toxic effects were reported. Yang and co-workers investigated GQDs-coated gold nanospheres, conjugated with FA and loaded with DOX, for photoacoustic and computed tomography imaging and as PTT agents [116]. When evaluated in vitro against HeLa and A549 cells, the nano-system demonstrated good biocompatibility and targeting ability towards cells with over-expressed FA receptors. Moreover, a strong reduction in cell viability was reported after laser irradiation, due to superior photothermal conversion effect of the nano-system. When injected through the caudal vein, in BALB/c nude mice inoculated with HeLa cell, the system reached the maximal accumulation at the tumor site after 8 h and, after irradiation, a consistent increase of temperature was recorded with consequent cancer cell death. In this study also, no weight loss was reported for the tested animals, thus demonstrating the absence of side effects and toxicity. A theranostic platform for synergistic cancer photoimmunotherapy and multimodal imaging, based on GQDs was synthesized by Liu and co-workers [117]. In this work, the authors conjugated amine functionalized GQDs with the PS Chlorin e6 (Ce6), coated the nano-system with polydopamine layers (PDOPA), and then assembled this photoactive nano-system with the immunostimulatory polycationic polymer/CpG oligodeoxynucleotide (CpG ODN) using Gd^3+^/Cy3 as imaging probes. The synthetic immuno-agent used in this study (CpG ODN) demonstrated the ability to target endosomal Toll-like receptor 9 (TLR9) leading to the activation and infiltration of T lymphocytes [124]. Studies in vitro using EMT6 murine mammary cancer cells, performed also under laser irradiation, demonstrated the effectiveness of combined photoimmunotherapy. The intratumor injection of the system in EMT6 tumor-bearing mice evidenced the tracking abilities of the system by high-quality bimodal magnetic resonance and fluorescence imaging and very limited systemic toxicity. Noteworthy, localized heating in the tumor region of mice was reported after 660 nm laser irradiation, monitored in real time by an IR thermal imaging camera. Ruiyi and co-workers investigated rare earth up-conversion NPs (UCNPs) and GQDs obtained from citric acid as nanoplatforms for pH and light-stimulated DOX release and chemo/PPT in vivo [118]. The cage-like nanostructure of the hybrid material constituted by GQDs and NaYF4:Yb,Tm nanocrystals and synthesized via coordination chemistry and hydrothermal treatment, was loaded with DOX, mixed with Au NPs and then covered with MGC-803 cell membranes to achieve homotypic targeting ability and to enhance the cellular uptake. The system demonstrated in vitro, against MGC-803 cancer cells, both pH and light-stimulated drug release as well as chemo/PTT with higher activity when compared with the free DOX. The authors investigated in vivo, in MGC-803 tumor-bearing Balb/c nude mice, the drug release, toxicity, NIR fluorescence imaging distribution and anticancer activity, reporting a higher accumulation of the drug inside the tumor tissues after NIR laser irradiation. An increase in temperature was observed inside cancer cells as a consequence of PTT effect with a volume reduction to 0.4% after 20 days from the intravenous injection. The authors also reported the absence of severe systemic toxicity to mice, since only mice treated with free DOX showed body weight loss. Wang and co-workers synthesized magnetic-induced GQDs for imaging-guided PTT in the second NIR window (1070 nm) via a one-step solvothermal treatment, starting from phenol as precursor [119]. The external magnetic field, with intensity of 9T, was demonstrated to improve the optical absorbance and the photoluminescence properties of the nanomaterials, due to the formation of superoxide radicals during the decomposition of phenol and leading to the formation of larger conjugated systems. The synthesized GQDs with a size of around 3.6 nm, demonstrated high photothermal conversion efficacy (33.45%). Biological experiments performed in vitro against 4T1 cells and in vivo in female BALB/c mice demonstrated the ability of the system to ablate tumor cells and inhibit the tumor growth under NIR-II irradiation. Enhanced NIR imaging of tumors in living mice was also reported. A smart nano-theranostic system based on nitrogen-doped GQDs (N-GQDs) for enhanced ultrasound/ fluorescence imaging and cooperative phototherapy was developed by Guan and co-workers [120]. The N-GQDs used in this study, 3 nm in size, were synthesized by a hydrothermal method starting from GO and DMF and coated with a mesoporous silica shell (HMSN). These spheres were then covered with a mesoporous carbon nitride (C_3_N_4_) layer and decorated with an amphipathic polymer (P-PEG-RGD) containing the hematoporphyrin photofrin (P) as photosensitizer, a PEG linker, and the tumor-homing peptide RGD (Arg-Gly-Asp) to achieve a targeted drug delivery to cancer cells over-expressing integrin α_v_β_3_ receptors [125]. The so assembled nano-system, named R-NCNP (Figure 5), was investigated in vitro in 4T1 cells, and in vivo in 4T1 tumor bearing Balb/c nude mice. The results of in vitro studies demonstrated the biocompatibility of the system and its ability to kill cancer cells after laser irradiation at 630 and 980 nm. In fact, upon irradiation, enhanced anticancer activity was detected, due to the oxygen generated by the C_3_N_4_-induced water splitting in the hypoxic tumor microenvironment. The generated O_2_ played also as echogenic source, making this system a laser-activatable ultrasound imaging agent. In addition, the PS maximized the yield of ROS, thus allowing for enhanced PDT. When injected in mice, the system demonstrated to actively target specific tumor tissues, leading to complete tumor ablation over 50 days. The so demonstrated PTT/PDT activity, assisted by triple-modal ultrasound (US), infrared thermal (IRT) and fluorescence (FL) imaging, was demonstrated to allow for the real-time monitoring of tumor ablation and therapy.

A simple system based on amine-functionalized GQDs (N-GQDs), prepared from citric acid and urea by a hydrothermal method, functionalized with the nucleus targeting TAT peptides and FA modified PEG, was exploited by Qi and coworkers for the treatment of solid tumors [103]. The system showed good biocompatibility, nucleus uptake, and cancer cell targeting ability when evaluated in Hela cells. Studies in vivo performed by intravenous administration of the system in HeLa tumor-bearing nude mice demonstrated optimal therapeutic outcomes against solid tumors, since no pathological alterations of the liver, kidney, or heart was reported. A nano-system based on GQDs embedded with mesoporous silica (MS@GQDs) for ultrahigh penetration and retention of solid tumor was reported by Srivastava and co-workers [121]. In this study, GQDs, synthesized from mango leaves by a microwave assisted method, were covered with mesoporous silica by sol-gel process and loaded with DOX. The system was then evaluated for NIR triggered drug delivery and anticancer activity, both in vitro in and in vivo. The in vitro studies performed against 4T1 cancer cells demonstrated the good biocompatibility of the system and the synergistic therapeutic response under NIR light exposure (800 nm). The intra-tumoral injection of the system in 4T1 tumor bearing Balb/c nude mice, followed by NIR light exposure, revealed a fast emission from the tumor area, which was retained up to a week. As a result of NIR irradiation, PTT activity was observed, with heat generation (∼52 °C) and tumor reduction of 68.75%, thus allowing the prolonged follow-up of image guided tumor regression. Branched polyethyleneimine (PEI) conjugated GQDs, functionalized with a fluorescent protein (GFP) with targeting ability towards cancer cells [126] and loaded with DOX, were developed by Lee and co-workers for the development of a pH-responsive nano-system for colon cancer treatment [122]. The presence of tertiary amines in the nano-system proved to exert a positive effect in the acidic microenvironment of the tumor, due to the high affinity with the negatively charged cell membrane, thus triggering drug release and cancer cell death. The authors reported good anticancer activity, both in vitro against HCT116 human colon cancer cell lines, and in vivo in a mice xenograft model, also demonstrating no toxicity in the animals used for this study. Recently, Zhou and coworkers have reported acidity-activated GQD-based nano-systems able to prolong the retention time of PSs in tumor tissues for imaging and PDT purposes [123]. In this study, small GQDs were transformed into larger systems (GQD NT) by conjugating three modules: (1) GQDs as nanoplatform able to load the PS tetra-carboxyl-phenyl porphyrin (TCPP) and the MRI contrast agent Mn-TCPP, through noncovalent interactions; (2) a targeting system constituted by RGD peptide; and (3) a linking module able to bind both GQDs and RGD through host–guest interactions between β-cyclodextrin and adamantine. The acidity of the tumor microenvironment allowed triggering of the transformation of the system and the drug release, while the PS molecule was transported to the tumor area by RGD mediated targeting. The transformed GQD NT was then disassembled over time, allowing the recovery of FLI and MRI signals, and the “turned-on” repeated PDT. The authors demonstrated in vitro, against different cell lines, the good targeting ability and the PDT effect of the GQT NT system. Studies in vivo, performed by intravenous administration of the system to A549 tumor-bearing mice, confirmed the benefits of the long tumor retention time for repeated PDT with only one injection, a marked tumor reduction after laser irradiation and good biocompatibility, since no detectable pathological changes in major organs were observed. The results of this study highlight the potential of GQD based systems as smart tools for next generation nanotherapeutics, exploiting promising approaches for long-time tumor imaging and effective solid tumor treatment.

## 3. Conclusions and Future Remarks

Nanomaterials with graphene-based structure represent, undoubtedly, ideal platforms for cancer detection and therapy, due to their outstanding chemical and physical properties, high surface area, unique possibilities of surface functionalization, and tunable biocompatibility. The presence of a graphene structure confers to all members of the GBN family high photothermal conversion efficiency under NIR light irradiation, thus opening the way to new strategies for the treatment of solid tumors. Nano-systems based on graphene, GO, rGO, and GQDs have been exploited for the development of nanotherapeutics for photoacoustic imaging-guided photothermal-chemotherapy and for photothermal and photodynamic therapy applied to tumor destruction. We have reviewed the more recent studies in vitro and in vivo related to GBN based theranostic tools, reporting outstanding results in terms of biocompatibility, cancer diagnosis and targeted therapy. It is expected that other advanced GBN-based systems will be developed in subsequent years with improved pharmacological and pharmacokinetic profile, less or no toxicity and high anticancer activity. However, despite all these achievements, several challenges need to be addressed before these fascinating tools can be employed for clinical use. Major issues related to the mechanisms of degradation in living systems, the route of diffusion across biological barriers and tissues, and the evaluation of long-term toxicity in different animal models still require further systematic investigation.

## Figures and Tables

**Figure 1 nanomaterials-13-02380-f001:**
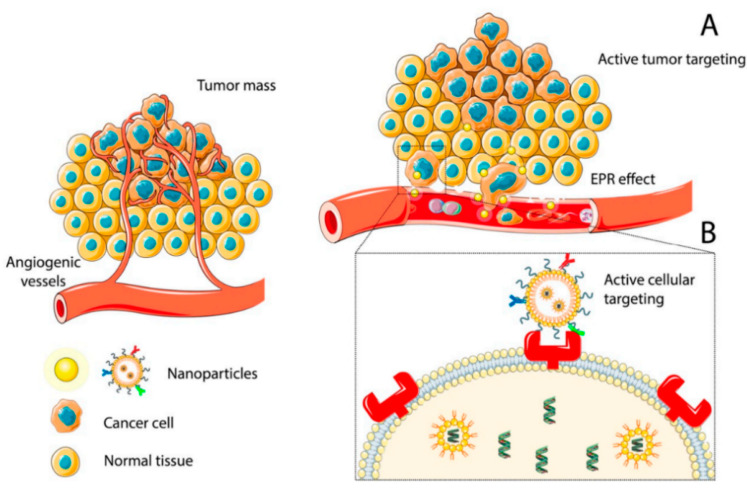
Accumulation of NPs at the tumor site by passive (**A**) or active (**B**) targeting. Reprinted from Ref. [14].

**Figure 2 nanomaterials-13-02380-f002:**
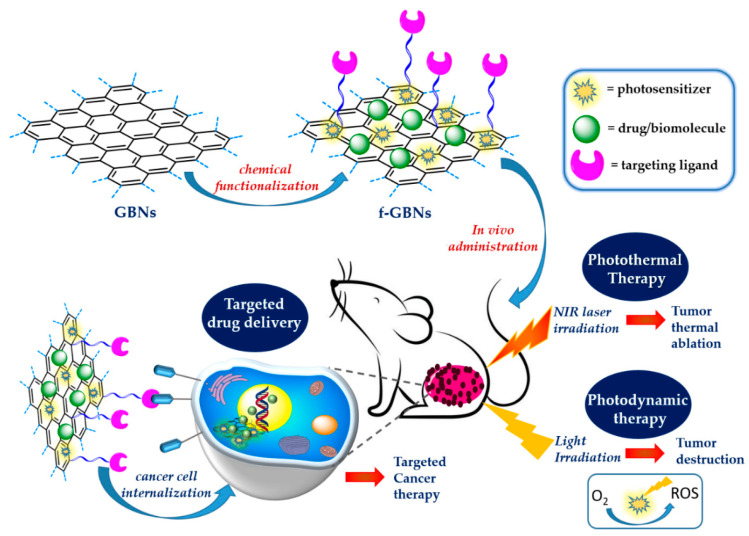
Surface functionalized 2D GBNs for solid tumors treatment.

**Figure 3 nanomaterials-13-02380-f003:**
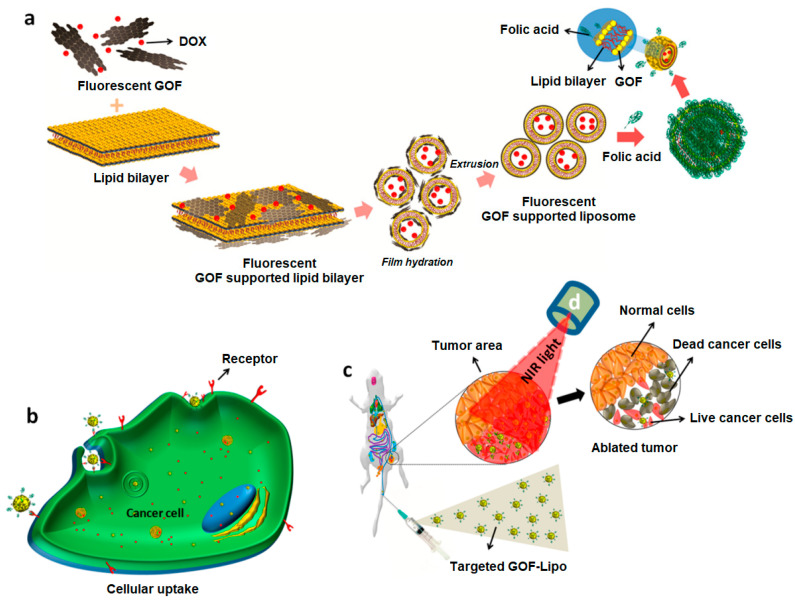
Schematic diagram of: (**a**) synthesis of DOX loaded and FA conjugated GOF-Lipo nanohybrid; (**b**) cell uptake of the targeted GOF-Lipo nanohybrid by receptor mediated endocytosis and drug distribution (red dots); (**c**) in vivo distribution and nanohybrid uptake via intravenous administration and NIR light mediated tumor regression. Reprinted with permission from Ref. [72]. Copyright (2019) American Chemical Society.

**Figure 4 nanomaterials-13-02380-f004:**
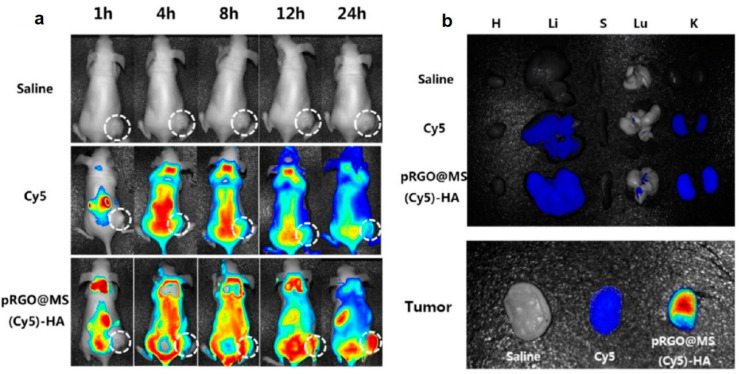
(**a**) In vivo fluorescence images of mice at different time points after administration of saline, Cy5, or pRGO@MS(Cy5)-HA (tumor pointed out with white circle). (**b**) Ex vivo fluorescence images of major organs and tumors at 24 h postinjection. Abbreviations are as follows: H, heart; Li, liver; S, spleen; Lu, lung; K, kidney. Reprinted with permission from Ref. [90]. Copyright (2017) American Chemical Society.

**Figure 5 nanomaterials-13-02380-f005:**
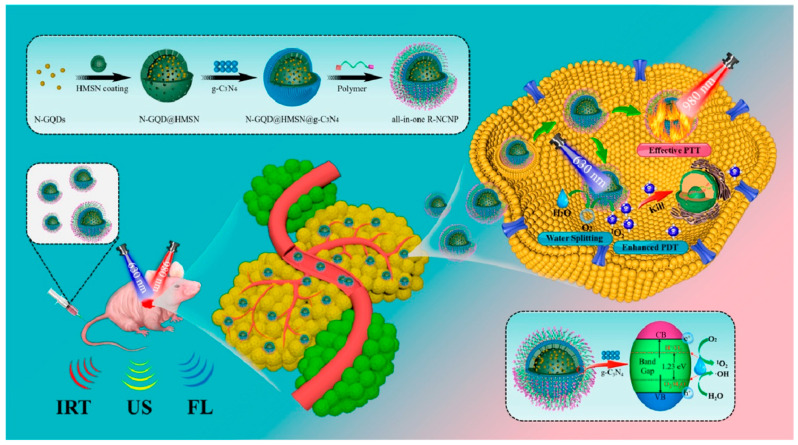
Schematic diagram of the synthesis route and PDT/PTT mechanism based on the R-NCNP nano-regulator. Reprinted with permission from Ref. [120]. Copyright (2020) American Chemical Society.

**Table 1 nanomaterials-13-02380-t001:** Graphene-based theranostic tools for solid tumors treatment.

Nano-System	Targeting Ligand	Drug	Photosensitizer/Labelling Agent	In Vivo Study	Ref.
GN-PEG-PPI	LHRH	-	Pc	xenografts of ovarian carcinoma bearing mice	[47]
GNFs	PSMA	(R)-Isp/DesB	^68^Ga	LNCaP tumor bearing mice	[48]
GNRs	Mannose/PRGD	-	-	xenografts of MDA-MB-231 tumor bearing mice	[49]

**Table 2 nanomaterials-13-02380-t002:** GO-based theranostic tools for solid tumors treatment.

Nano-System	Targeting Ligand	Drug	Photosensitizer/Labelling Agent	In Vivo Study	Ref.
GO-CS	MPG	miRNA	-	A375 tumor bearing nude mice	[67]
SiPc@GO	-	-	Pc	xenograft of MCF-7 tumor bearing mice	[68]
γ-Fe_2_O_3_@GO-PEG	-	DOX	γ-Fe_2_O_3_	H22 tumor bearing nude mice	[69]
MnWO_4_@GO-PEG	-	DOX	MnWO_4_	4T1 tumor bearing mice	[70]
Pd@GO	-	Pd	-	PC3 tumor bearing BALB/c nude mice	[71]
GOF-Lipo	FA	DOX	-	4T1 tumor bearing Balb/c mice	[72]
GO-PDA-BSA	FA	DTPA-Mn(II)	5-Fu	Wistar rats	[73]
SPIONs@GO	-	DOX	SPIONs	4T1 tumor-bearing mice	[74]
GO-PEG	EGFR	5-Fu	-	CRC tumor-bearing BALB/c mice	[75]

**Table 3 nanomaterials-13-02380-t003:** rGO-based theranostic tools for solid tumor treatment.

Nano-System	Targeting Ligand	Drug	Photosensitizer/Labelling Agent	In Vivo Study	Ref.
ICG-PDA-rGO	-	-	ICG	4T1 tumor-bearing mice	[88]
Au NRs@rGO-PEG	Tat protein	-	Cy7	U87MG tumor-bearing mice	[89]
PDA-rGO@MS	HA	DOX	Cy5	HeLa tumor bearing BALB/c mice	[90]
rGO	-	-	-	Panc02-H7 tumor-bearing C57BL/6 mice	[91]
Fe_3_O_4_/rGO-PEG	-	-	-	4T1 tumor-bearing BALB/c mice	[92]
rGO-PEG	FA	epacadostat	Cy7	CT26 tumor bearing BALB/c mouse	[93]
rGO/Ag	-	-	-	EAC tumor-bearing mice	[94]

**Table 4 nanomaterials-13-02380-t004:** GQDs-based theranostic tools for solid tumors’ treatment.

Nano-System	Targeting Ligand	Drug	Photosensitizer/Labelling Agent	In Vivo Study	Ref.
RBC@GQDs-NS	Ct	DTX	-	A549 nude mice	[115]
Au@GQDs	FA	DOX	-	HeLa tumor-bearing BALB/c nude mice	[116]
PDOPA@GQDs	ODN	-	Ce6, Gd^3+^	EMT6 tumor-bearing mice	[117]
Au@GQDs	MGC-803 cell membranes	DOX	NaYF4:Yb,Tm	MGC-803 tumor bearing Balb/c nude mice	[118]
9T-GQDs	-	-	-	BALB/c mice	[119]
R-NCNP	RGD	-	P	4T1 tumor bearing Balb/c nude mice	[120]
N-GQDs	TAT/FA	-	-	HeLa tumor-bearing BALB/c nude mice	[103]
MS@GQDs	-	DOX	-	4T1 tumor bearing Balb/c nude mice	[121]
PEI@GQDs	GPC	DOX	-	HCT116 tumor bearing nude mice	[122]
GQDs	RGD	-	TCPP/Mn-TCPP	A549 tumor-bearing mice	[123]

## Data Availability

Not applicable.

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
