# Peer review of "Theranostic Applications of 2D Graphene-Based Materials for Solid Tumors Treatment"

_nanomaterials, 2023, doi:10.3390/nano13162380_

Round 1

Reviewer 1 Report (Previous Reviewer 4)

From a scientific point of view just like last time I found it a well-documented review with recent bibliography, well-structured and discussed in some way from the authors' own perspective.

In the current version significant improvements have been made, based on the comments of the reviewers.

I still think it is a quality review that should be given a chance.

Author Response

We really thank the Reviewer for having appreciated our work and for his positive comments.

Reviewer 2 Report (New Reviewer)

The manuscript has basic rational structure, but I noticed there’s still some flaws in it, please revise them before publication:

1.     Please explain the source of Figure 2

2.     Line 99 and line 112 are repeated for the “NIR” short name.

3.     For ref 30, I could see any statement about “NIR light in the range of 700–1000 nm” content in that paper, I'm worried that the order of your references is garbled, please re-check carefully.

4.     A single paragraph should be listed to state the commonalities and differences (chemistry and physical properties) of graphene, GO, rGO, and GQDs, and explain why they have different applications.

5.     I recommend adding a list of acronyms at the end of the manuscript.

Some minor grammar issues such as the use of definite and indefinite articles

Author Response

This manuscript is a resubmission of an earlier submission. The following is a list of the peer review reports and author responses from that submission.

Round 1

Reviewer 1 Report

The manuscript provides an overview of the latest advancements in the synthesis of nanosystems based on graphene, graphene oxide (GO), and graphene quantum dots (GQDs). This timely review can be accepted in Nanomaterials with minor revisions.

1. The abstract part is a little short, which should be a high summary of the whole content. It can supplement some harm of cancer to human society at the beginning, and then introduce the materials and treatment methods.

2. Other treatments can be added in the introduction, such as microwave thermotherapy, microwave dynamic therapy, ultrasonic thermotherapy, ultrasound thermal and dynamic therapy, and highlights the advantages of photothermal/photodynamic therapy by comparing treatment modalities.

3. The cases cited in this review have not been comprehensively analyzed, and should be analyzed from preparation method, structure, anti-tumor performance, biocompatibility and other aspects. Please add.

4. Please correct the numerous typos in the manuscript.

The manuscript provides an overview of the latest advancements in the synthesis of nanosystems based on graphene, graphene oxide (GO), and graphene quantum dots (GQDs). This timely review can be accepted in Nanomaterials with minor revisions.

1. The abstract part is a little short, which should be a high summary of the whole content. It can supplement some harm of cancer to human society at the beginning, and then introduce the materials and treatment methods.

2. Other treatments can be added in the introduction, such as microwave thermotherapy, microwave dynamic therapy, ultrasonic thermotherapy, ultrasound thermal and dynamic therapy, and highlights the advantages of photothermal/photodynamic therapy by comparing treatment modalities.

3. The cases cited in this review have not been comprehensively analyzed, and should be analyzed from preparation method, structure, anti-tumor performance, biocompatibility and other aspects. Please add.

4. Please correct the numerous typos in the manuscript.

Reviewer 2 Report

The manuscript is not particularly informative and well organized to provide an insightful discussion in this topic. Furthermore, there are only three images included in the article, and they do not align with the corresponding references. Additionally, numerous errors are present within the manuscript. For instance, while there are 76 references listed, the text refers to 104 references.

None

Reviewer 3 Report

The review presents an overview of 2D graphene-based materials for cancer theranostic applications, however this topic has been already broadly reported in the literature for many years and many reviews have been published recently, some examples of which are listed here:

Mahmoudifard M. Graphene family in cancer therapy: recent progress in cancer gene/drug delivery applications[J]. Journal of Materials Chemistry B, 2023.

Wang Y, Li J, Li X, et al. Graphene-based nanomaterials for cancer therapy and anti-infections[J]. Bioactive materials, 2022, 14: 335-349.

Bhatt H N, Pena-Zacarias J, Beaven E, et al. Potential and progress of 2D materials in photomedicine for cancer treatment[J]. ACS applied bio materials, 2023, 6(2): 365-383.

Laraba S R, Luo W, Rezzoug A, et al. Graphene-based composites for biomedical applications[J]. Green Chemistry Letters and Reviews, 2022, 15(3): 724-748.

Chakraborty A R, Akshay R, Sahoo S, et al. Emerging graphene-based nanomaterials for cancer nanotheranostics[M]//Handbook of Porous Carbon Materials. Singapore: Springer Nature Singapore, 2023: 1091-1126.

Abd Elkodous M, Olojede S O, Sahoo S, et al. Recent advances in modification of novel carbon-based composites: Synthesis, properties, and biotechnological/biomedical applications[J]. Chemico-Biological Interactions, 2023: 110517.

Patil S, Rajkuberan C, Sagadevan S. Recent biomedical advancements in graphene oxide and future perspectives[J]. Journal of Drug Delivery Science and Technology, 2023: 104737.

Jafari A, Chenab K K, Malektaj H, et al. An attempt of stimuli-responsive drug delivery of graphene-based nanomaterial through biological obstacles of tumor[J]. FlatChem, 2022, 34: 100381.

Shevalkar G B, Prajapati M K, Mali K. A brief overview on theranostic applications of graphene and graphene-based nanomaterials[J]. Nanomaterial-Based Drug Delivery Systems: Therapeutic and Theranostic Applications, 2023: 295-325.

The authors miss to state the urgency and innovative contribution of their review and lack a comprehensive description of the topic, e.g., graphene-based materials are now also used in sonodynamic cancer therapy, nanozyme catalytic therapy, and other areas.

So I judge the present manuscript as a meager repetition of what was already stated previously with no new insight and for this reason, I recommend to do not publish it.

Reviewer 4 Report

            In the Review articles “Theranostic applications of 2D graphene-based materials for solid tumors treatment” the author reviewed the most recent advances related to the  2D graphene-based nanomaterials (GBNs) and their application for cancer therapy.

·         In the first part of the introduction the authors provide an overview of the topic addressed, namely the use of 2D graphene (functionalized graphene, graphene oxide (GO), reduced GO (rGO) and graphene quantum dots (GQDs), small 2D graphene fragments) and their application in drug delivery, biosensing, bioimaging, and photothermal therapy. The advantages of their use in solid tumor layering are described in detail.

·         Various types of surface functionalization with different combinations are described and exemplified with relatively recent bibliographies. Figure 1 shows visually, very expressively all these types of functionalization. Is Figure 1 original, or is there a copyright agreement for it?

·         The advantages of using 2D GBNs in the treatment of solid tumors are described in detail, highlighting the minimum characteristics that these materials must fulfill for safe and effective application. In the following the three most important categories of 2D GBNs (Graphene, Graphene Oxide and Graphene quantum Dots) are discussed separately, as well as their application in the mentioned application.

·         The authors managed to make a very clear classification of the three categories of materials, specifying each time the advantages and disadvantages they have in the application of solid tumors treatment. Examples are given of the various specific functions with which graphene has been functionalized, the type of testing to which it has been subjected and the effects of treatment, compared with other alternatives.

·         In Table 1 shows Graphene-based theranostic tools for solid tumors treatment. I personally believe that in this table more than three systems should be mentioned (because they exist in the literature) for Graphene type 2D GBNs, so that the reader has a better overview of the use of these specific systems in the treatment of solid tumors.

·         The next two sub-categories of 2D GBNs are very well described, Graphene Oxide and Graphene quantum Dots, exemplified and the treatment mechanism well explained.

·         The description is very well condensed, logical and easy to follow and the information provided is efficient and of very good quality. At the end of the chapter, the reader understands the types of materials, the advantages and disadvantages of each, as well as their mechanisms of action in the treatment of solid tumors.

·         This review article is an excellent material that presents in a clear and concise way the whole field of applications of 2D GBNs materials in solid tumor cancer treatment. It represents a very good state of the art of the field.

·         The conclusions are succinct and well argued, well pointed and support the original purpose of the article